# SIaM: Self-Improving Code-Assisted Mathematical Reasoning of Large Language Models

## Abstract

There is a growing trend of teaching large language models (LLMs) to solve mathematical problems through coding. Existing studies primarily focus on prompting powerful, closed-source models to generate seed training data followed by in-domain data augmentation, equipping LLMs with considerable capabilities for code-assisted mathematical reasoning. However, continually training these models on augmented data derived from a few datasets such as GSM8K may impair their generalization abilities and restrict their effectiveness to limited question types. Conversely, the potential of improving such LLMs by leveraging large-scale, expert-written, diverse math question-answer pairs remains unexplored. To utilize these resources and tackle unique challenges such as code response assessment, we propose a novel paradigm that uses a code-based critic model to guide steps including question-code data construction, quality control, and complementary evaluation. We also explore different alignment algorithms with self-generated instruction/preference data to foster continuous self-improvement. Experiments across both in-distribution (up to $+5.7\%$) and out-of-distribution ($+4.4\%$) benchmarks in English and Chinese show the effectiveness of the proposed paradigm.

## 1 Introduction

Though large language models (LLMs) have demonstrated strong performance on mathematical benchmarks, they still face challenges in achieving accurate computation and reasoning, especially in out-of-distribution scenarios. For example, even the recent closed-source LLM o1-mini struggles with multiplication beyond eight digits (Deng, 2024) using step-by-step reasoning (or Chain-of-Thought, CoT) (Wei et al., 2022). To alleviate the computational burden on LLMs, particularly those of smaller sizes, there is a growing trend of utilizing code and code interpreters to enhance precise computation and reasoning of LLMs in solving mathematical problems (Chen et al., 2022; Gao et al., 2023b; Zhou et al., 2023). An effective method involves prompting closed-source LLMs to generate code-based solutions for given questions. However, previous studies demonstrated that closed-source models, without extra test-time compute, still struggle with real-world high school and college-level math exams (Liu et al., 2024). Solving advanced problems through coding demands not only mathematical expertise but also interdisciplinary knowledge and skills, including programming and natural language, making it a more formidable challenge. Previous code-assisted studies primarily focus on using closed-source LLMs such as GPT-4 to label a few small-scale, representative datasets such as GSM8K (Cobbe et al., 2021) and MATH (Hendrycks et al., 2021), verifying the correctness of the solutions via pattern-based answer matching, and training models on the verified data for further **in-distribution** data augmentation through sampling, code execution, and answer validation (Wang et al., 2023; Liu et al., 2023; Gou et al., 2024; Lu et al., 2024). However, continually learning from these datasets or their augmented versions, regardless of the use of code, is evidently less effective for improving the generalization of LLMs due to the limited diversity.

On the other hand, large-scale, expert-written, mathematical question-answer (QA) pairs from educational web resources remain under-studied to improve code-assisted math reasoning abilities of LLMs. These resources span educational levels from primary school to college and include various question types and answer formats, such as multiple-choice, application, proof, and cloze. To use these resources to self-improve code-assisted[1] LLMs, instead of further extensively distilling

---

[1]Using the data to compare CoT with code-assisted reasoning or enhancing CoT is not the focus of this work.

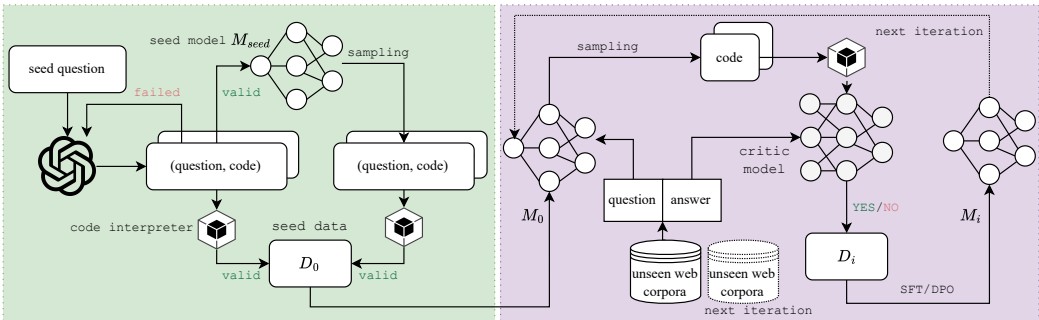

Figure 1: Overview of our self-improving code-assisted paradigm using large-scale web QA data.

closed-source models, one natural solution is to use a fine-tuned model to generate code samples for each problem and use the valid data to (iteratively) improve this LLM, similar to self-improved CoT reasoners (Zelikman et al., 2022; Yuan et al., 2023; Xu et al., 2024; Hosseini et al., 2024) over data with reference answers. However, **the key challenge is to determine whether the self-generated code responses align with reference answers in diverse formats**. Fortunately, with the aid of an external code interpreter, we are less concerned about potential computation errors in intermediate CoT reasoning steps. We assume a code solution is more likely to be correct if its execution result matches the reference answers, thus shifting the focus from the step-by-step comparison to comparing the reference answers with the code execution results. Based on our analysis (Section 3.1), we observe that most cases primarily require format conversion between plain text and code syntax (e.g., "(x-5)(x^2-4x+7)" vs. "(x-5)*(x**2-4*x+7)" and "(1, -2, 2, -3)" vs. "{A:1, B:-2, C:2, D:-3}") and relatively simple numerical calculations, which do not require advanced logical reasoning abilities or in-depth language-specific knowledge (Section 3.5).

These observations and task simplification motivate us to design a critic model to evaluate the correctness of the code execution result against the reference answer by simply predicting YES or NO (see examples in Table 1). As illustrated in Figure 1, this critic model is used to guide multiple steps during self-improvement. We first train a model with seed question-code data following previous code-assisted studies and consider it as the initial policy model. In each iteration, we use the current policy model to generate code samples for new questions and keep the highest-scoring valid code responses rated by the critic model for supervised fine-tuning (SFT) in the subsequent iteration. To foster continuous improvement, we also explore different preference learning algorithms such as DPO (Rafailov et al., 2024) and ORPO (Hong et al., 2024) with self-generated preference data, where the preference labels are also provided by the critic model.

We perform experiments on various model families, such as Llama3-8B (AI@Meta, 2024) and DeepSeek-Coder-7B (Daya Guo, 2024), and Qwen2-7B (Yang et al., 2024). Experimental results across both in-distribution (up to $+5.7\%$) and out-of-distribution (OOD) $(+4.4\%)$ benchmarks in English and Chinese show the effectiveness of self-improving LLMs using our proposed paradigm with large-scale mathematical QA pairs. The resulting 7-8B models can outperform state-of-the-art 70B code-assisted math LLMs (Gou et al., 2024) by $11.9\%$ in OOD scenarios. Notably, we observe a strong correlation between the traditional heuristic-based evaluation method and the critic model (Section 3.5), with the latter reducing the additional human effort needed to design rules for new mathematical benchmarks. Additionally, introducing SFT loss into the DPO training is surprisingly effective in controlling the code response length. To summarize the contributions of this work:

- To the best of our knowledge, this is the first attempt to leverage large-scale web QA pairs to improve the code-assisted mathematical reasoning abilities of LLMs.
- To better leverage these large-scale, diverse QA pairs, we propose a novel iterative self-improving paradigm that employs a new critic model to guide various steps such as data construction and filtering. This critic model can also serve as a complementary evaluation scorer, reducing the reliance on heuristic design for new evaluation tasks.
- Extensive experiments on both English and Chinese tasks demonstrate the effectiveness of our paradigm, and our comprehensive analysis of the key factors in achieving continuous improvement at different stages may shed light on future studies.

Table 1: Example instances of the code-based critic task ((question, reference answer) pairs come from the web QA data, and Chinese texts are translated into English).

| | |
|---|---|
| **System Prompt** | Your goal is to evaluate whether the candidate answer provided by the model for a math problem matches the reference answer. Here are the steps to complete the task:
– First, carefully read the given math problem.
– Next, review the reference answer for the math problem.
– Then, examine the candidate answer provided by the model, which may include a program and the result of running that program.
– Finally, summarize whether the candidate answer matches the reference answer or can be made to match through simple calculations/conversions.
– The response format should be Yes or No. |
| **Instruction** | ### Question\n\n Given $f(1 - 2x) = 3x + 1$, find $f(-3)=$ __ ·
### Reference Answer\n \n 7 \n \n
### Candidate Answer\n \n from sympy import symbols, Eq, solve\n# Define the variable\n x = symbols('x')\n# Establish the equation based on the problem conditions\n equation = Eq(1 - 2*x, -3)\n# Solve the equation\n solution = solve(equation, x, dict=True)\n solution_value = solution[0][x]\n # Calculate f(-3)\n f_neg_3 = 3*solution_value + 1\n f_neg_3<solution>7</solution>\n\n
### Assessment |
| **Output** | Yes |
| **Instruction** | ### Question\n\n Given the sequence $\{a_n\}$ with the sum of the first $n$ terms $S_n = 2n - 3$, what is $a_3$? A: 3 B: 2 C: 1 D: 0
### Reference Answer\n \n B \n \n
### Candidate Answer\n \n # Define a function to calculate the nth term\n def calculate_nth_term(n):\n # Based on the derived formula\n an = 2 ** n - 2 ** (n - 1)\n return an\n # Calculate the value of the 3rd term \n a3 = calculate_nth_term(3) # a3 <solution>4</solution>\n\n
### Assessment |
| **Output** | No |

## 2 METHOD

### 2.1 TRAINING AN INITIAL MODEL

One key factor for an iterative paradigm is to have a reasonably strong initial model. To train such a model, we first use high-quality seed data to fine-tune an LLM, resulting in model $M_{\text{seed}}$. We use $M_{\text{seed}}$ to generate code samples and keep up to four predictions per question wherein the execution result of the code matches the reference answer and combines the seed data and the self-distilled data to train $M_0$, which is further used as the initial model for later stages. We will introduce more details about the seed data construction in the experiment section.

### 2.2 BUILDING A MULTI-USE CODE-BASED CRITIC MODEL

To improve LLMs with large-scale, diverse-format math QA data without code annotations, several challenges arise in data utilization, filtering, and evaluation. First, previous studies primarily use pattern-based methods to compare predictions and ground truth answers during validation and evaluation. This works well for GSM-style datasets, where answers are single numbers and well-formatted (e.g., *"72"* in *"...72 clips altogether in April and May.\n #### 72"*). However, pattern-based methods face inherent challenges in handling diverse answer types and formats and bridging the gap between natural language and programming language. For example, with the MATH dataset, comparing CoT predictions with reference answers in LaTeX-like format already requires human-written patterns and answer conversion (Yue et al., 2023). This complexity is compounded when predictions are presented in code syntax, even when the critic task is already simplified to compare the reference answer with the code execution result.

To address the above challenges, we propose building a code-based critic model optimized by the following objective:

$$L(r_\phi) = -\log r_\phi(y \mid q, a, c, e), \tag{1}$$

where $q$ denotes a question, $a$ is the reference answer to $q$, $c$ represents the code response to $q$, and $e$ is the execution result of code $c$. To simplify the task, we let $y$ be either "YES" or "NO". Examples are shown in Table 1. We leave other formulations, such as training a scalar critic model (Ouyang et al., 2022), to future work.

## 2.3 CODE DATA GENERATION

As mentioned previously, our goal is to leverage web math QA data to continuously self-improve the code-assisted mathematical reasoning ability of LLMs. For well-formatted, web-collected math data such as APE (Zhao et al., 2020) and CM (Qin et al., 2021), where most answers are one or two numerical values (see examples in Table 16), it is efficient and effective to compare the reference answer and the execution result of the code using scripts released by previous studies (Section 3.2). For real-world math data involving various types of questions, such as multiple-choice, multi-part questions, fill-in-the-blank, application, and proof, using a critic model introduced in the previous section is more flexible and saves the need for the labor-intensive and time-consuming process of creating task-specific patterns. Note that for all questions, we only use their reference answers to verify the correctness of code execution results, rather than directly using these answers – often short and inconsistent in style – for training. Additionally, we only use benchmarks' training sets.

In the $k + 1$-th iteration, for each new question, we use the current policy model $\pi_{\theta_k}$ to generate five code samples and execute them to obtain the results. For questions in the diverse-format web data, the critic model is then used to predict YES or NO for each response $(a_i, c_{ij}, e_{ij})$ given $q_i$. We use the probability of YES or NO as the confidence value for the critic model's judgment. A higher probability score indicates a greater confidence in the code response, either agreeing with or disagreeing with the reference answer.

## 2.4 SELF-IMPROVEMENT WITH UNSEEN DATA

One natural choice is to perform supervised fine-tuning (SFT) on $\pi_{\theta_k}$ using $D_{\text{SFT}}$:

$$L_{\text{SFT}}(\pi_{\theta_{k+1}}) = -\log \pi_{\theta_{k+1}}(c \mid q) \tag{2}$$

$$D_{\text{SFT}} = \{(q_i, c_{ij}) \mid r_\phi(y = \text{YES} \mid q_i, a_i, c_{ij}, e_{ij})\} \tag{3}$$

As critics may contain errors, we explore using the probability of each judgement (i.e., YES or NO) as a confidence score to filter out noise. Besides, we introduce extra constraints: for each question, we only retain the highest-scoring positive instance $t_{ij} = \{q_i, a_i, c_{ij}, e_{ij}\}$, similar to rejection (Bai et al., 2022) or Best-of-N sampling (Stiennon et al., 2020), where $t_{ij} \in T_i$ of the same question $q_i$. To encourage models to learn from more challenging problems, if all instances in $T_i$ are judged as YES, we discard this question and its corresponding generated code from consideration.

$$
\begin{aligned}
D_{\text{SFT, H}} = \{(q_i, c_{ij}) \mid & \; r_\phi(y = \text{YES} \mid t_{ij}), \\
& \; p_{r_\phi}(y = \text{YES} \mid t_{ij}) > \lambda_1, \\
& \; t_{ij} = \arg\max_{t_{ij} \in T_i} p_{r_\phi}(y = \text{YES} \mid t_{ij}), \\
& \; \sum_{j=1}^{|T_i|} \mathbf{1}\{r_\phi(y = \text{NO} \mid t_{ij})\} \geq \lambda_2\}
\end{aligned}
\tag{4}
$$

where $\lambda_1, \lambda_2$ represent thresholds for filtering and difficulty control.

In addition to supervised fine-tuning a policy model on self-generated SFT data ($D_{\text{SFT, H}}$ or $D_{\text{SFT}}$), we further leverage negative instances by optimizing the policy model on preference data using algorithms such as DPO (Rafailov et al., 2024) and ORPO (Hong et al., 2024). Compared to SFT, these preference learning algorithms additionally decrease the probability of losing responses. We mainly focus on DPO and leave other options for future studies, and we jointly train the policy with the SFT objective to alleviate overfitting to the preference data and ensure a stable update (Hong et al., 2024). See more discussions on the impact of the SFT objective, especially its role in controlling the response length, in Section 3.4.

$$L_{\text{DPO}}(\pi_{\theta_{k+1}}) = -\log \sigma \left( \beta \log \frac{\pi_{\theta_{k+1}}(y_w \mid x)}{\pi_{\theta_k}(y_w \mid x)} - \beta \log \frac{\pi_{\theta_{k+1}}(y_l \mid x)}{\pi_{\theta_k}(y_l \mid x)} \right) - \lambda \cdot \log \pi_{\theta_{k+1}}(y_w \mid x) \tag{5}$$

We can easily leverage our critic model to build preference $(c_w, c_l)$ pairs, where $c_w$ represents the winning code and $c_l$ represents the losing code. For each question, we use the highest-scoring YES response and the highest-scoring NO response to form a preference "best-and-worst" pair, aiming to maximize the difference between them. See preference data examples in Section A.6.

$$
\begin{aligned}
D_{\text{DPO}} = \{(q_i, c_{ij}, c_{ik}) \mid & r_\phi(y = \text{YES} \mid t_{ij}), \\
& r_\phi(y = \text{NO} \mid t_{ik}), \\
& t_{ij} = \arg \max_{t_{ij} \in T_i} p_{r_\phi}(y = \text{YES} \mid t_{ij}), \\
& t_{ik} = \arg \max_{t_{ik} \in T_i} p_{r_\phi}(y = \text{NO} \mid t_{ik})\}
\end{aligned}
\tag{6}
$$

## 3 EXPERIMENTS

### 3.1 DATA

We summarize the statistics of data used for self-improvement in Table 2 and evaluation benchmarks in Table 15. The large-scale in-house math QA pairs (1.13M in total) are used in compliance with the authorized licenses from educational websites. It covers various educational stages from primary school to college and question types like prof, application, cloze, and multiple-choice questions (e.g., questions in Table 1). More examples (e.g., questions in Table 18 and Table 19) and analysis of the web QA data can be found in the Appendix. While we will not release the full QA pairs, we will release our code, seed data in English, self-improved/critic models, and self-generated SFT/preference data to facilitate future studies.

**Seed Data** $D_0$: To generate the seed data for English, following previous work, we use GPT-4-0613 to generate Python code in an iterative fashion: we repeatedly sample the remaining questions that do not have correct code (i.e., the code execution results match the reference answer of the questions) for up to three iterations. We use questions from the training sets of GSM8K (7.5K) and MATH (7.5K) as the seed questions for imitation learning. For datasets such as GSM8K in which the answers are mostly single numbers, it is easier to compare answer and code execution results. After two iterations, we can annotate $98.5\%$ of questions in GSM8K. For datasets such as MATH wherein the answers

Table 2: Statistics of training data used in our three-stage paradigm ($D_1$ and $D_{2,\text{in-house}}$ are Chinese resources; $D_{2,\text{WebInstruct}}$ is English-dominant).

| Data/Subset | | QA Source | Size |
|---|---|---|---|
| $D_0$ | zh | web | 76K |
| | en | GSM8K, MATH | 44K |
| $D_1$ | | APE, CM | 211K |
| $D_{2,\text{in-house}}$ | SFT | | 893K |
| | SFT(H) | educational websites | 273K |
| | DPO | | 465K |
| $D_{2,\text{WebInstruct}}$ | DPO | pre-training corpora | 447K |

are diverse in formats, we simply keep the code that can be successfully executed without errors. For seed questions for Chinese, we randomly sample 20K math questions from the in-house web QA data and follow the same procedure using GPT-4-0613 for code generation to construct the Chinese subset of $D_0$. For each question, we add a language-specific system prompt: *"Please write a python code to solve the following questions"* or its Chinese counterpart, "请用python代码解决以下问题".

**Value-Style** $D_1$: We utilize the initial policy $M_0$ to generate code samples to questions in training sets of two open-source word math problem datasets APE (200.5K) (Zhao et al., 2020) and CM (13.6K) (Qin et al., 2021), both collected from educational websites covering elementary and middle-school levels. Since all the answers are one or two numerical values, for efficiency, we use heuristics with Python to compare the code execution results with reference answers for validation. We keep up to four valid code samples for each question.

**Diverse-Format Data** $D_2$ **and Critic Data**: To increase the diversity of our training data, we further consider large-scale mathematical QA pairs (excluding those used for seed data) mentioned previously. For each question, we retain only one positive code and one negative code (if any exists) judged by the critic. To better understand this web data and the critic task, we analyze the reference answers for 50 instances. Only $14\%$ of them are single numerical values, while $50\%$ involve format conversion (e.g., syntax or structure) when the answers are expressions, equations, coordinates, sets, etc. Another difference between real-world data and well-formatted benchmarks is the inconsistency

in the format of reference answers. Specifically, half of the answers contain CoT-style (Wei et al., 2022) explanations and/or irrelevant contents, such as tags and URLs, while the rest are in a short form. This makes it challenging to use this QA data directly to improve CoT reasoning or to parse short-form answers for easier verification with a few patterns, as done for clean benchmarks (e.g., answer indicators "###" for GSM8K and "BOXED{ }" for MATH). For multiple-choice or multi-part questions ($8\%$ in total), we additionally require the question context for mapping option labels and their contents, as well as question decomposition. These observations reflect the diversity of question types in our web QA data. See statistics in the Appendix (Table 13).

To evaluate the generalization and robustness of our paradigm, we also use a recently released large-scale reasoning QA dataset named WebInstruct (Yue et al., 2024) to construct a similar-scale $D_2$, containing 447K preference pairs (see examples in Section A.6). Compared to our in-house web QA data, WebInstruct is mostly in English and is extracted from the pre-training corpora. Therefore, the answers are not guaranteed to be written by educational experts as our Chinese web data.

To build the training data for the critic model, we use $M_0$ to generate code samples for randomly sampled questions from $D_2$ and execute these code samples. We then prompt GPT-4-0613 with the input (question, code, code result, reference answer) following the template in Table 1. After filtering, we retain 16.8K training instances, of which $48.6\%$ of are judged as YES.

## 3.2 IMPLEMENTATION

We use LLLAMAFACTORY (Zheng et al., 2024) for efficient fine-tuning built upon DeepSpeed (ZeRO-3). Our experiments are conducted using 8XA100 40GB GPUs. We train LLMs with BF16 mixed-precision. The training for the self-improving paradigm takes approximately 96 hours. With 80 workers in multi-processing mode on a CPU machine, we can execute about 9,003 code samples per minute. Each model at each stage is trained for two epochs with a learning rate of 1e-5 for SFT and 1e-6 for preference learning. We set the SFT loss coefficient ($\lambda$ in Equation 7) to $1.0$. The maximum sequence length is set to $1024$, and the batch size is set to $64$. We set $\lambda_1$ to $0.8$ and $\lambda_2$ to $3$.

We experiment with various LLMs to select backbone models such as CodeLlama-7B-Python (Roziere et al., 2023), Llama3$_{\text{instruct}}$ (AI@Meta, 2024), CodeQwen1.5-7B-Chat (Team, 2024), QWEN2(Yang et al., 2024), and Deepseek-Coder-7B-instruct-v1.5 (Daya Guo, 2024), which demonstrate strong coding capabilities on code-related benchmarks. Due to limited computational resources, we use their 7-8B versions with their default templates and leave the model scaling up for future work. We primarily follow the evaluation scripts from previous studies (Liang et al., 2024) for Chinese benchmarks and FastEval[2] for English benchmarks GSM8K and MATH, which often use Python for numerical comparison. We also make adjustments to these scripts, as our predicted answers are in code syntax. CodeLlama-7B-Python is used as the backbone model to train the code-based critic model for three epochs with the maximum sequence length $4096$.

## 3.3 THE PERFORMANCE OF THE INITIAL POLICY AND SELF-IMPROVED LLMS

As shown in Table 3, we experiment with three backbone models for self-improvement — DeepSeek$_{\text{code}}$, Llama3$_{\text{instruct}}$, QWEN2Math$_{\text{instruct}}$ — that show superior average performance across math datasets in both Chinese (APE, CM, and CMATH (Wei et al., 2023)) and English (GSM8K and MATH) than other investigated models when trained with seed data (see complete results of initial policy models based on eight LLMs in Table 9). Therefore, we consider them as initial policy models (i.e., M$_0$) for self-improvement. After two additional iterations on the unseen data D$_1$, and D$_2$ constructed with the help of our code-based critic model, the resulting models (i.e., M$_2$) consistently outperform M$_0$ by a large margin on Chinese benchmarks.

We observe that self-improving the initial policy model with **Chinese-only** data, D$_1$ and D$_2$, does not hurt the accuracy of M$_2$ on English tasks. In fact, it may be beneficial (e.g., $+1.5\%$ on both MATH and GSM8K datasets using DeepSeek$_{\text{code}}$). Conversely, adding English seed data ($36.7\%$ of D$_0$) consistently improves M$_0$'s average performance on Chinese benchmarks (D$_0$ vs. $D_{0,\text{zh}}$ in Table 4). To some extent, we may interpret code as a universal language for solving mathematical problems across different languages. The language-specific parts are mainly in the code comments, which are relatively indirect for problem-solving via code execution. Thus, our paradigm may reduce the

---

[2]`github.com/FastEval/FastEval`.

Table 3: Accuracy across the dev sets of math datasets. All Chinese datasets are OOD for $M_0$. CMATH is OOD for $M_2$ as the training sets of CM and CMATH are later used for distant supervision.

| Model | Size (B) | Tool | Chinese Tasks | | | English Tasks | |
|---|---|---|---|---|---|---|---|
| | | | CM | APE | CMATH | GSM8K | MATH |
| GPT-4-1106-Preview | – | ✗ | – | 84.2 | 89.3 | 93.6 | 53.6 |
| Qwen-Chat (Bai et al., 2023) | 72 | ✗ | – | 77.1 | 88.1 | 76.4 | 31.8 |
| ChatGLM-Math (Xu et al., 2024) | 32 | ✗ | – | 89.4 | 85.6 | 82.6 | 40.6 |
| Skywork-Math (Yang et al., 2023) | 13 | ✗ | – | 74.4 | 77.3 | 72.3 | 17.0 |
| Math-InternLM2 (Team, 2023) | 20 | ✗ | – | 75.2 | 78.5 | 82.6 | 37.7 |
| MetaMath (Yu et al., 2023a) | 70 | ✗ | – | – | – | 82.3 | 26.6 |
| MathCoder (Wang et al., 2023) | 34 | ✓ | – | – | – | 81.7 | 45.2 |
| ToRA (Gou et al., 2024) | 70 | ✓ | – | – | – | 84.3 | 49.7 |
| | 7 | ✓ | – | – | – | 72.6 | 44.6 |
| MathGenieLM (Lu et al., 2024) | 70 | ✓ | – | – | – | 88.4 | 51.2 |
| MinT (Liang et al., 2024) | 7 | ✓ | 77.6 | 76.0 | – | 40.8 | – |
| **Initial Model Baselines ($M_0$)** | | | | | | | |
| QWEN2Math$_{instruct}$ | 7 | ✓ | 84.9 | 83.4 | 87.3 | 79.5 | 48.0 |
| DeepSeek$_{code}$ | 7 | ✓ | 82.7 | 81.2 | 87.0 | 77.4 | 44.4 |
| Llama3$_{instruct}$ | 8 | ✓ | 83.3 | 83.2 | 87.2 | 76.8 | 41.8 |
| **Self-Improvement with Chinese Diverse-Format Web Data ($M_2$)** | | | | | | | |
| SIaM(QWEN2Math$_{instruct}$) | 7 | ✓ | 90.1 (+5.2) | 88.1 (+4.7) | 93.2 (+5.9) | 81.5 (+2.0) | 50.0 (+2.0) |
| SIaM(DeepSeek$_{code}$) | 7 | ✓ | 87.3 (+4.6) | 85.9 (+4.7) | 91.2 (+4.2) | 78.9 (+1.5) | 45.9 (+1.5) |
| SIaM(Llama3$_{instruct}$) | 8 | ✓ | 89.0 (+5.7) | 86.8 (+3.6) | 90.8 (+3.6) | 80.5 (+3.7) | 41.9 (+0.1) |

burden of preparing large-scale, language-specific math data for each language. We observe similar trends on DeepSeek$_{code}$ and QWEN2Math$_{instruct}$, as shown in Table 4.

We list several general-purpose/math-specified multi-lingual/English LLMs for reference. Note that direct comparisons are challenging due to differences in architectures, pre-training corpora, alignment algorithms, model size, the use of tools, and labeled data. For example, code-assisted methods ToRA, MathCoder, and MathGenieLM are trained on 69K, 80K, and 170K English-only data, respectively, augmented based on GSM8K and MATH. In contrast, our experiments use 44K English seed data and explore the use of large-scale Chinese math QA pairs. Moreover, the evaluation scripts, originally designed for plain-text answers instead of code outputs, may cause an underestimation of our methods' performance on datasets such as MATH, where answers involve more expressions and structures beyond numerical values. This also highlights the need for a more flexible evaluation method.

Table 4: Impacts of different stages and data selection on the development sets of datasets.

| Model | Stages | Data | CM | APE | CMATH | GSM8K | MATH | Average |
|---|---|---|---|---|---|---|---|---|
| QWEN2Math$_{instruct}$ | SFT | $D_{0,en}$ | – | – | – | 78.5 | 47.7 | – |
| | SFT | $D_{0,zh}$ | 83.9 | 83.8 | 87.0 | – | – | – |
| | SFT | $D_0$ | 84.9 | 83.4 | 87.3 | 79.5 | 48.0 | 76.6 |
| | SFT | $D_0 + D_1$ | 87.8 | 85.9 | 88.3 | 79.2 | 49.5 | 78.1 |
| | SFT $\rightarrow$ DPO | $D_0 + D_1$; $D_{2,WebInstruct}$ | 87.8 | 86.0 | 88.5 | 82.4 | 48.7 | 78.7 |
| | SFT $\rightarrow$ DPO | $D_0 + D_1$; $D_{2,in-house}$ | **90.1** | **88.1** | **93.2** | **81.5** | **50.0** | **80.6** |
| DeepSeek$_{code}$ | SFT | $D_{0,en}$ | – | – | – | 74.6 | 43.8 | – |
| | SFT | $D_{0,zh}$ | 81.0 | 82.4 | 86.8 | – | – | – |
| | SFT | $D_0$ | 82.7 | 81.2 | 87.0 | 77.4 | 44.4 | 74.5 |
| | SFT | $D_0 + D_1$ | 87.0 | 84.3 | 88.0 | 77.6 | 44.6 | 76.3 |
| | SFT $\rightarrow$ DPO | $D_0 + D_1$; $D_{2,WebInstruct}$ | 87.0 | 84.4 | 88.2 | 78.2 | 44.4 | 76.5 |
| | SFT $\rightarrow$ DPO | $D_0 + D_1$; $D_{2,in-house}$ | **87.3** | **85.9** | **91.2** | **78.9** | **45.9** | **77.8** |
| Llama3$_{instruct}$ | SFT | $D_{0,en}$ | – | – | – | 75.1 | 37.2 | – |
| | SFT | $D_{0,zh}$ | 82.5 | 83.3 | 85.5 | – | – | – |
| | SFT | $D_0$ | 83.3 | 83.2 | 87.2 | 76.8 | 41.8 | 74.4 |
| | SFT | $D_0 + D_1$ | 87.6 | 85.0 | 89.0 | 76.6 | 41.8 | 76.0 |
| | SFT $\rightarrow$ DPO | $D_0 + D_1$; $D_{2,WebInstruct}$ | 87.5 | 86.1 | 88.7 | 80.2 | **42.1** | 76.9 |
| | SFT $\rightarrow$ DPO | $D_0 + D_1$; $D_{2,in-house}$ | **89.0** | **86.8** | **90.8** | **80.5** | 41.9 | **77.8** |

## 3.4 THE COMPARISON OF DIFFERENT CHOICES OF DATA AND ALIGNMENT METHODS

**Diversity & Quality**: Based on the experimental results, given $D_0$ and $D_1$, we observe that two-stage SFT (first on $D_0$ for two epochs and then on $D_1$ for two epochs) under-performs one-stage SFT

(over the concatenation of $D_0$ and $D_1$ for two epochs) (B vs. C in Table 5). However, incorporating $D_2$ using either strategy achieves similar performance (E vs. F in Table 5). One possible reason may be that the questions in $D_1$ are from two web-collected value-style benchmarks (APE and CM), resulting in less diversity compared with $D_2$, which has a broader range of question types (Section 3.1). Ensuring the diversity of data in each stage may help the model generalize better across various types of math questions, similar to the observations seen when training general-purpose LLMs (e.g., (Shen et al., 2023)).

As mentioned previously, we use the code-based critic model to construct SFT data. Since the process will inevitably introduce false positive data, we further consider several constraints for filtering (Equation 4 in Section 2.4). Experimental results show that we can achieve similar average accuracy using either $D_{2,\text{SFT,H}}$ or the $D_{2,\text{SFT}}$ (D vs. E in Table 5). However, $D_{2,\text{SFT,H}}$ is only 30.6% of the latter's size, indicating the usefulness of the filtering process.

Table 5: The average accuracy of Llama3$_{\text{instruct}}$ on the dev sets of five datasets after alignment.

| ID | Alignment | Data | Accuracy |
|----|-----------|------|----------|
| A | SFT | $D_0$ | 74.4 |
| B | SFT $\to$ SFT | $D_0$ ; $D_1$ | 75.4 |
| C | SFT | $D_0 + D_1$ | 76.0 |
| D | SFT | $D_0 + D_1 + D_{2,\text{SFT}}$ | 76.1 |
| E | SFT | $D_0 + D_1 + D_{2,\text{SFT,H}}$ | 76.1 |
| F | SFT $\to$ SFT | $D_0 + D_1$ ; $D_{2,\text{SFT,H}}$ | 76.2 |
| G | SFT $\to$ SFT | $D_0 + D_1$ ; $D_{2,\text{DPO, winning}}$ | 76.0 |
| H | SFT $\to$ ORPO | $D_0 + D_1$ ; $D_{2,\text{DPO}}$ | 77.0 |
| I | SFT $\to$ DPO | $D_0 + D_1$ ; $D_{2,\text{DPO}}$ | 77.8 |

**DPO or SFT**: Based on a reasonably good model $M_1$ (trained with $D_0$ and $D_1$, such as C in Table 5), we can either self-improve it via SFT or DPO (Section 2.4). We compare using the (question, winning code) pairs in the DPO data for another round of SFT, which results in a 1.8% drop in accuracy on downstream tasks (G vs. I in Table 5). Since we do not impose strict constraints on the winning code responses in DPO, $D_{2,\text{DPO, winning}}$ is 1.7 times the size of $D_{2,\text{SFT,H}}$. Still, using the filtered SFT data $D_{2,\text{SFT,H}}$ achieves slightly better performance (F vs. G), showing the effectiveness of filtering.

**DPO with SFT**: Our experiments indicate that DPO training is relatively insensitive to the weight ($\lambda$ in Equation 7) of the SFT loss. We tested with $\lambda = 1.0$ and $\lambda = 2.0$, both of which resulted in similarly good performance (77.8%). However, as shown in Table 6, removing the SFT loss (i.e., $\lambda = 0$) from DPO training leads to a dramatic increase in response length, especially for Chinese tasks such as CMATH, and yields worse results than the reference policy model (C in Table 5). This observation aligns with discussions on length exploitation issue of the original DPO loss (Park et al., 2024). One possible reason for the length control achieved by adding the SFT loss could be that the winning responses used for the SFT loss are generated

Table 6: The impact of the weight of the SFT loss in DPO training on the average accuracy and average response length in words on GSM8K and CMATH ($L_0$: response length of reference policy).

| $\lambda$ | GSM8K | | | CMATH | | |
|-----------|-------|---|-------------|-------|---|-------------|
| | ACC | L | $\frac{L}{L_0}$ | ACC | L | $\frac{L}{L_0}$ |
| reference model | | | | | | |
| - | 76.6 | 323 | 1.0 | 89.0 | 136 | 1.0 |
| 0.0 | 73.4 | 1834 | 5.7 | 57.5 | 3160 | 23.2 |
| 0.5 | 78.8 | 532 | 1.6 | 90.7 | 201 | 1.5 |
| 1.0 | 80.5 | 352 | 1.1 | 90.8 | 136 | 1.0 |
| 1.5 | 79.0 | 328 | 1.0 | 90.7 | 135 | 1.0 |
| 2.0 | 79.8 | 326 | 1.0 | 90.7 | 134 | 1.0 |

by the reference policy model. By setting a larger weight to SFT, we control the deviation from the reference policy, which alleviates a substantial increase in response length. We also experiment with using ORPO (Hong et al., 2024), which removes the need for a reference model and jointly trains with the SFT loss. However, this method is not as effective as jointly training DPO and SFT in our experiments on Llama3$_{\text{instruct}}$ (H vs. I in Table 5) and the other two backbone models (Table 17).

**Other Diverse-Format Resources**: We also experiment with constructing similar-scale preference data using the diverse-format $D_2$ based on WebInstruct (Section 3.1). However, the resulting improvement in average accuracy is less substantial compared to that achieved with the Chinese diverse-format $D_2$ (+0.9% vs. +1.8% on Llama3$_{\text{instruct}}$ in Table 4; +0.6% vs. +2.5% on QWEN2Math$_{\text{instruct}}$ in Table 4). One possible reason for this difference could be that the QA pairs in the WebInstruct extracted from pre-training corpora, despite being of similar scale used for experiments, may provide weaker

supervision compared to those sourced from educational websites, where answers are typically written by experts. Although we have filtered out QA pairs where reference answers contain no numbers, we observe that some questions still do not require any calculations as they are originally collected for improving the **general reasoning** abilities of LLMs, such as *"How is the interquartile range (IQR) connected to percentiles?"* or related to other subjects such as *"What is the most prevalent state of matter in the universe ...?"*, while our mathematical benchmarks for evaluation primarily require numerical computation. Nevertheless, these results demonstrate the robustness of our paradigm.

## 3.5 Using the Critic Model as a Complementary Evaluator

We have shown the effectiveness of using the critic model to construct SFT and preference data, and all scores are computed by comparing predictions with ground truth answers, using heuristics-based exact match (EM) following previous studies for fair comparisons. To explore the potential of using the critic model as a complementary evaluator, we examine the correlation between the two evaluation methods on the previously used benchmarks. We use the critic model to compare the code execution result and the original ground truth answers (final-step answers if answers are COT-style) (e.g., "3750", "[12, 18]", and "$\backslash\backslash\texttt{frac}\{1\}\{2\}$") in these benchmarks. Since all scores are either 0

Table 7: Correlation of two evaluation methods: heuristics-based EM and the critic model. ACC represents the average accuracy of our best-performing $M_2$ on downstream tasks **rated** by the two methods on downstream tasks.

| Dataset | Correlation$_{\text{Kendall}}$ | ACC$_{\text{EM}}$ | ACC$_{\text{critic}}$ |
|---|---|---|---|
| CM | 0.66 | 89.0 | 84.6 |
| APE | 0.76 | 86.8 | 86.5 |
| CMATH | 0.77 | 90.8 | 91.8 |
| GSM8K | 0.97 | 80.5 | 80.6 |
| MATH | 0.79 | 41.9 | 48.2 |
| average | 0.79 | 77.8 | 78.3 |

(No) or 1 (Yes), we report the Kendall's $\tau$ between the two methods. As shown in Table 7, **there is a very strong correlation** (0.79) (compared to the very-strong-cutoff value 0.71 and strong-cutoff value 0.49 (Schober et al., 2018)) between the scores computed by the two evaluators. The strong associations in English tasks are surprising, given that the critic model is trained on Chinese-only data. This may be due to (i) the backbone model being a well-instructed model focused on English, and (ii) comparing answers to mathematical questions relying less on language-specific knowledge.

## 3.6 The Performance of Self-Improved LLMs on More Out-of-Distribution Tasks

Considering the above results in Section 3.5, we are now more confident in using the critic model to evaluate models' performance on additional OOD benchmarks, without the need to write extensive heuristics for different tasks. Besides CMATH, we evaluate the OOD performance of our models using Math-Bench (Liu et al., 2024), a math benchmark supporting evaluation in both Chinese and English. The open-ended or multiple-choice questions in MathBench span various educational stages, from primary school to college levels. We report scores on its two subsets: MathBench-A, which evaluates practical problem-solving skills, and MathBench-T, which assesses theoretical understanding.

Table 8: OOD accuracy on MathBench ($\star$: scored by the critic model; $\dagger$: based on the numbers reported by (Liu et al., 2024); A: application; T: theoretical).

| model | Tool | Subset-A | Subset-T | ACC$_{\text{average}}$ |
|---|---|---|---|---|
| GPT-4-0125-Preview | $\times$ | 58.8$^\dagger$ | 78.4$^\dagger$ | 68.6$^\dagger$ |
| GLM4 | $\times$ | 51.3$^\dagger$ | 73.1$^\dagger$ | 62.2$^\dagger$ |
| Qwen-Chat-72B | $\times$ | 49.7$^\dagger$ | 77.2$^\dagger$ | 63.5$^\dagger$ |
| Math-InternLM2-20B | $\times$ | 41.9$^\dagger$ | 64.3$^\dagger$ | 53.1$^\dagger$ |
| Llama3$_{\text{instruct}}$-8B | $\times$ | 36.7$^\dagger$ | 52.1$^\dagger$ | 44.4$^\dagger$ |
| MathCoder-7B | $\checkmark$ | 32.6$^\star$ | 27.4$^\star$ | 30.0$^\star$ |
| MathCoder-34B | $\checkmark$ | 50.1$^\star$ | 49.3$^\star$ | 49.7$^\star$ |
| ToRA-7B | $\checkmark$ | 31.0$^\star$ | 28.4$^\star$ | 29.7$^\star$ |
| ToRA-70B | $\checkmark$ | 54.3$^\star$ | 54.4$^\star$ | 54.3$^\star$ |
| **Language-specific system prompt:** | | | | |
| SIaM(Llama3$_{\text{instruct}}$)$_0$-8B | $\checkmark$ | 62.5$^\star$ | 57.9$^\star$ | 60.2$^\star$ |
| SIaM(Llama3$_{\text{instruct}}$)$_2$-8B | $\checkmark$ | 66.7$^\star$ | 62.6$^\star$ | 64.6$^\star$ |
| **Chinese-only system prompt:** | | | | |
| SIaM(Llama3$_{\text{instruct}}$)$_0$-8B | $\checkmark$ | 64.0$^\star$ | 64.4$^\star$ | 64.2$^\star$ |
| SIaM(Llama3$_{\text{instruct}}$)$_2$-8B | $\checkmark$ | 69.5$^\star$ | 65.8$^\star$ | 67.6$^\star$ |

As shown in Table 8, the self-improved models demonstrate substantial gains on both subsets, with an accuracy improvement of 4.4%. On both subsets, the self-improved model consistently outperforms the initial one across all educational levels with notable improvements, particularly in answering middle school and English theoretical questions. See sub-category performance in Tables 10 and 11 (Section A.3). Note that we provide the scores of other CoT models

for reference, as they are judged by a different scorer. We compare our method with ToRA and MathCoder, two strong code-aided math LLMs, rated by the same critic model. Although trained on English-only data, ToRA-70B and MathCoder-34B demonstrates reasonable performance on Chinese tasks. Nevertheless, our 8B model also outperforms the best-performing ToRA-70B on the English subset of MathBench by 14.5% and 9.2%, respectively, on A and T (Table 12). In addition, we observe that our self-improved model performs better when the Chinese system prompt is applied to solve English questions. This may be due to the fact that our training data primarily consists of Chinese data with Chinese system prompts.

Compared to practical application questions, it seems that using CoT, LLMs are much better at handling theoretical knowledge questions. In contrast, solving all questions via coding shows balanced and reasonable performance. This demonstrates the advantage of using tools to aid in computation but also indicates the limitations of relying solely on code to address questions that may not require actual computation. It remains an open question whether, and how, code can be used to assist advanced theoretical reasoning (Liu et al., 2024)–a topic beyond the scope of this paper.

## 4 RELATED WORK

**Evaluation**: For automatic math evaluation on well-formatted benchmarks, previous studies mostly use heuristics and external tools (e.g., the Python EVAL() function) to compare answers and predictions (Fourrier et al., 2023; Gao et al., 2023a), which works quite well for single numerical value answers, as seen in datasets such as GSM8K (Cobbe et al., 2021), ASDiv (Miao et al., 2020), and SVAMP (Patel et al., 2021). However, since answers from web resources are diverse in formats and language-code syntactic differences, using carefully designed task-specific heuristics becomes less feasible for comparing answers and code execution results. For datasets beyond value-style answers such as MATH (Hendrycks et al., 2021), closed source LLMs are also used for evaluation such as OpenAI-Evals. However, this approach is not cost-effective for assessing large-scale code samples.

**Self-Improvement**: Several approaches (Li et al., 2023; Yu et al., 2023b; Lu et al., 2023; Yuan et al., 2024; Hu et al., 2024) use the LLM itself or a separate critic model (Ouyang et al., 2022) for scoring or validating natural-language responses. This work focuses on tool-assisted assessment of code responses to math questions. Similar to previous self-improvement CoT studies (Zelikman et al., 2022; Hosseini et al., 2024; Yuan et al., 2023; Xu et al., 2024), we use ground truth answers to assist training data validation and filtering, as it is still challenging to train a good critic/reward model for math reasoning without reference answers, even for solution-level assessment (Lightman et al., 2023; Daheim et al., 2024). In our paradigm, a single iteration of DPO can already enhance performance, and additional iterations on unseen data might further improve results, as suggested by previous online studies with CoT reasoning (Dong et al., 2024; Zhang et al., 2024). However, it has been shown that using a general-purpose reward model yields fewer improvements in mathematical reasoning compared to the gains observed in other tasks.

**Data Augmentation and Knowledge Distillation**: Though recent studies have shown that CoT or code-assisted in-distribution data augmentation will lead LLMs to achieve strong performance on in-distribution math datasets (Luo et al., 2023; Yu et al., 2023a; An et al., 2023; Li et al., 2024), we leave data augmentation on web data (either CoT or code-assisted reasoning) for future work. We only use GPT-4 to annotate seed/critic training data, and using closed-source LLMs to annotate the code responses of large-scale web questions is not explored. Unfortunately, SOTA code-aided models (Gou et al., 2024; Wang et al., 2023), trained on English data, have limited capability in labeling diverse-format questions written in Chinese.

## 5 CONCLUSIONS AND FUTURE WORK

We introduce a novel paradigm for improving LLMs, which employs a code-based critic model to guide stages such as the creation and filtering of question-code data as well as complementary evaluation. We also investigate various alignment algorithms using self-generated instruction/preference data for further improvement. Results show the effectiveness of self-improving LLMs with this proposed paradigm. Future research includes studying post-training on code-only data to enhance the computational capabilities of LLMs and improvement of the critic model.

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

# A APPENDICES

## A.1 BACKBONE COMPARISONS FOR INITIAL MODEL SELECTION

Although QWEN2 also demonstrates strong performance, we use its math-specific variant to ensure the diversity of selected backbone models. For the same reason, and given the marginal performance difference between Llama3$_{\text{instruct}}$ and Llama3$_{\text{base}}$ when both are fine-tuned on $D_0$, we only Llama3$_{\text{instruct}}$ for our experiments.

## A.2 IMPACTS OF STAGES AND DATA SELECTION

Ablation studies of the stages and data selection on the development sets of datasets.

## A.3 SUB-TYPE PERFORMANCE ON MATHBENCH

The data presented in the tables clearly shows the advantage of SIaM(Llama3$_{\text{instruct}}$)$_2$ over SIaM(Llama3instruct)$_0$ across various educational levels. For both the MathBench-A and MathBench-T datasets, SIaM(Llama3instruct)$_2$ consistently outperforms SIaM(Llama3instruct)$_0$. In the MathBench-A dataset, improvements are seen in all levels from Primary to College, with notable jumps in Middle and High school levels (6.7% and 7.0% improvement, respectively, in Table 10).

Table 9: Accuracy across the development sets of math datasets of initial policy models based on different backbone models.

| Model | Size (B) | Tool | Chinese Tasks | | | English Tasks | | Average |
| | | | CM | APE | CMATH | GSM8K | MATH | |
|---|---|---|---|---|---|---|---|---|
| CodeLlama | 7 | ✓ | 77.7 | 78.0 | 84.5 | 69.7 | 37.6 | 69.5 |
| QWEN$_{code}$ | 7 | ✓ | 81.9 | 81.5 | 86.0 | 71.9 | 41.4 | 72.6 |
| Llama3.1$_{instruct}$ | 8 | ✓ | 82.4 | 82.1 | 86.2 | 76.5 | 41.1 | 73.6 |
| Llama3$_{base}$ | 8 | ✓ | 83.9 | 82.6 | 86.8 | 76.8 | 41.9 | 74.4 |
| Llama3$_{instruct}$ | 8 | ✓ | 83.3 | 83.2 | 87.2 | 76.8 | 41.8 | 74.4 |
| DeepSeek$_{code}$ | 7 | ✓ | 82.7 | 81.2 | 87.0 | 77.4 | 44.4 | 74.5 |
| QWEN2 | 7 | ✓ | 83.9 | 82.8 | 87.3 | 77.7 | 44.4 | 75.2 |
| QWEN2Math$_{instruct}$ | 7 | ✓ | 84.9 | 83.4 | 87.3 | 79.5 | 48.0 | 76.6 |

Similarly, the MathBench-T dataset shows improvement across all levels, particularly in the Middle school and English categories, which demonstrate $8.1\%$ and $10.5\%$ increases, respectively. These results indicate that SIaM(Llama3instruct)$_2$ provides enhanced accuracy in out-of-distribution scenarios, making it a more reliable choice for varied educational levels.

In the seed data $D_0$, we use a language-specific system prompt for each English instance: *"Please write a python code to solve the following questions"*. For the Chinese subset of $D_0$ and all instances in $D_1$ and $D_2$ — which are exclusively Chinese data — we use a consistent Chinese system prompt "请用python代码解决以下问题" (*"Please write a python code to solve the following questions"*). When evaluating our self-improved model on MathBench, we observe that it performs better when the Chinese system prompt is applied to solve English questions (Table 11). This may be due to the fact that our training data primarily consists of Chinese data with Chinese system prompts.

We compare with SOTA code-assisted models trained on augmented MATH and GSM8K datasets ToRA and MathCoder. Before detailed comparisons, we first review the background of the use of code for mathematical reasoning. Code can be used either directly (Chen et al., 2022; Gao et al., 2023b) (code-only) or interactively (Wang et al., 2023) during problem-solving. The latter approaches such as ToRA and MathCoder jointly solve problems using CoT explanation and code. One advantage of these interactive methods over code-only methods is that the final step of their solution is usually written in CoT, allowing the easy use of existing scripts designed for CoT-style benchmarks for evaluation. However, this does not allow for robust comparisons for unseen diverse-format comparisons. In addition, the role of using tools multiple times to address a single math problem is unclear based on the performance difference of interactive methods (Table 3). For example, ToRA needs 1.02 tool interaction rounds per question while MathCoder requires 2.05 for MATH and GSM8K. This work focuses on the direct usage of code as a case study to avoid multi-step inference and leave the interactive setting for future studies.

For ToRA 7B[3] and 70B[4] models, we use their official inference scripts.[5] On MathBench, ToRA needs an average of 1.00 and 1.01 tool interaction rounds per question. It seems its final CoT reasoning primarily focuses on adjusting formatting answers to fully leverage existing CoT evaluation scripts. We use ToRA's generated code and execution result, keeping the rest of the inputs for the critic model the same. We also experiment with replacing the execution results with the CoT outputs, but this does not result in significant changes. Our self-improved 8B model outperforms one SOTA code-assisted model, ToRA-70B, across all subcategories on this OOD dataset (Table 12).

For MathCoder, we evaluate its best-performing 34B model[6] and 7B model[7], which needs 1.53 and 2.13 tool interaction rounds per question, respectively. We also use their released inference scripts[8] and follow the data format.

Table 10: Fine-grained OOD accuracy on the MathBench dataset scored by the critic model using language-specific system prompts.

| Level | MathBench-A | | MathBench-T | |
|---|---|---|---|---|
| | $\text{SIaM(Llama3}_{\text{instruct}})_0$ | $\text{SIaM(Llama3}_{\text{instruct}})_2$ | $\text{SIaM(Llama3}_{\text{instruct}})_0$ | $\text{SIaM(Llama3}_{\text{instruct}})_2$ |
| **Arith** | 98.0 | **99.0** | – | – |
| **Primary** | 75.7 | **80.7** | 66.6 | **67.5** |
| **Middle** | 56.3 | **63.0** | 60.1 | **68.2** |
| **High** | 50.3 | **57.3** | 59.1 | **60.6** |
| **College** | 32.0 | **33.3** | 50.2 | **57.9** |
| **Chinese** | 56.8 | **63.5** | 62.7 | **63.6** |
| **English** | 66.2 | **68.8** | 50.6 | **61.1** |

Table 11: Fine-grained OOD accuracy on the MathBench dataset scored by the critic model using a Chinese-only system prompt.

| Level | MathBench-A | | MathBench-T | |
|---|---|---|---|---|
| | $\text{SIaM(Llama3}_{\text{instruct}})_0$ | $\text{SIaM(Llama3}_{\text{instruct}})_2$ | $\text{SIaM(Llama3}_{\text{instruct}})_0$ | $\text{SIaM(Llama3}_{\text{instruct}})_2$ |
| **Arith** | 97.3 | **98.3** | – | – |
| **Primary** | 71.0 | **79.0** | **71.6** | 71.3 |
| **Middle** | 60.0 | **69.0** | 70.3 | **71.5** |
| **High** | 54.0 | **59.7** | 61.8 | **62.3** |
| **College** | 37.7 | **41.3** | 59.2 | **62.6** |
| **Chinese** | 57.3 | **63.7** | **63.8** | 63.4 |
| **English** | 68.4 | **73.3** | 65.4 | **69.5** |

Table 12: Fine-grained OOD accuracy of ToRA (70B and 7B) and MathCoder (34B and 7B) on the MathBench dataset scored by the critic model (T: ToRA; M: MathCoder).

| Level | MathBench-A | | | | MathBench-T | | | |
|---|---|---|---|---|---|---|---|---|
| | T-7B | T-70B | M-7B | M-34B | T-7B | T-70B | M-7B | M-34B |
| **Arith** | 39.3 | **82.7** | 40.7 | 66.3 | – | – | – | – |
| **Primary** | 40.3 | **77.7** | 43.3 | 70.0 | 30.9 | **53.9** | 26.2 | 47.0 |
| **Middle** | 24.3 | 39.7 | 28.7 | **45.3** | 31.0 | **57.6** | 25.0 | 46.8 |
| **High** | 30.0 | **39.7** | 29.7 | 39.0 | 28.0 | **51.9** | 28.2 | 47.8 |
| **College** | 21.0 | **31.7** | 20.7 | 30.0 | 25.5 | **55.1** | 29.0 | 54.3 |
| **Chinese** | 28.2 | **47.5** | 23.0 | 41.5 | 26.5 | **50.5** | 25.4 | 43.8 |
| **English** | 32.9 | **58.8** | 39.0 | 55.9 | 31.2 | **60.3** | 30.4 | 57.5 |

## A.4 DATA STATISTICS

We only use GPT-4 to generate seed question-code training data: 76K for Chinese and 44K for English (Table 2). This scale is similar to those (CoT or code-assisted) in previous work (e.g., (Wang et al., 2023; Gou et al., 2024; Luo et al., 2023)) for a single language. See Table **??** for comparisons. The output of the critic model is simply "Yes" or "No", which is a much cheaper labeling task compared to traditional generation tasks.

## A.5 OTHER ALIGNMENT ALGORITHMS

As shown in Table 17, DPO demonstrates superior performance compared to ORPO, both with the SFT loss. We leave the exploration of more length-regularized alignment algorithms and the role of the reference policy model in preference optimization to future studies.

---

[3]https://huggingface.co/llm-agents/tora-code-7b-v1.0.

[4]https://huggingface.co/llm-agents/tora-70b-v1.0.

[5]https://github.com/microsoft/ToRA/tree/main.

[6]https://huggingface.co/MathLLMs/MathCoder-CL-34B.

[7]https://huggingface.co/MathLLMs/MathCoder-CL-7B.

[8]https://github.com/mathllm/MathCoder.

Table 13: Distribution of different types in the dataset.

| Type | Percentage (%) |
|------|----------------|
| Noisy step-by-step rationale | 50 |
| Numerical value | 30 |
| Expression | 20 |
| Set | 20 |
| Equation | 8 |
| Multiple-choice | 6 |
| Multi-questions | 2 |
| Coordinates | 2 |
| Other | 14 |

Table 14: Overview of datasets and their labelers, languages, and scales.

| Dataset | Labeler | Tool | Language | Scale |
|---------|---------|------|----------|-------|
| WizardMath (Luo et al., 2023) | ChatGPT | × | en | 96K |
| MetaMath (Yu et al., 2023a) | GPT-3.5-Turbo | × | en | 395K |
| MathCoder (Wang et al., 2023) | GPT-4 | ✓ | en | 49K |
| ToRA (Gou et al., 2024) | GPT-4 | ✓ | en | 16K |
| $D_0$ (Ours) | GPT-4 | ✓ | en | 44K |
| $D_0$ (Ours) | GPT-4 | ✓ | zh | 76K |

Table 15: Statistics of evaluation benchmarks. Note that in our experiments, we do not use any rationale in these datasets as we focus on solving problems via coding. We only use the questions and short-form answers from the training set of MATH and GSM8K for constructing the seed data, and we use the questions and short-form answer from the training set of APE and CM for constructing the data for self-improvement.

| Dataset | Language | Answer Type | Level | Training | Validation |
|---------|----------|-------------|-------|----------|------------|
| APE (Zhao et al., 2020) | zh | numerical value | elementary | 200,488 | 5,000 |
| CM (Qin et al., 2021) | zh | numerical value(s) | grades 6—12 | 13,628 | 1,703 |
| CMATH (Wei et al., 2023) | zh | numerical value | elementary | – | 600 |
| MathBench (Liu et al., 2024) | en, zh | mixed | from primary to college | – | 3,709 |
| MATH (Hendrycks et al., 2021) | en | mixed | college | 7,500 | 5,000 |
| GSM8K (Cobbe et al., 2021) | en | numerical value | elementary | 7,473 | 1,319 |

Table 16: An example instance of the APE dataset (Zhao et al., 2020) (we translate the question into English; ⋆: we do not use this rationale in our paradigm).

| | |
|---|---|
| **Question**: | Given: Apples cost 6 yuan for 4 kilograms, and oranges cost 11 yuan for 5 kilograms. Uncle Wang buys 16 kilograms of apples and 20 kilograms of oranges. How much should he pay in total? |
| **Answer**: | 68 |
| **Rationale⋆**: | x=6/4*16+11/5*20 |

Table 17: The self-improving performance in different stages on the development sets of different datasets. The best open-sourced performance for each backbone model is highlighted in bold.

| Model | Alignment | Data | CM | APE | CMATH | GSM8K | MATH | ACC$_{average}$ |
|-------|-----------|------|-----|-----|-------|-------|------|----------------|
| DeepSeek$_{code}$ | SFT | $D_0 + D_1$ | 87.0 | 84.3 | 88.0 | 77.6 | 44.6 | 76.3 |
| | SFT → ORPO | $D_0 + D_1; D_2$ | **87.7** | 85.5 | **91.2** | 76.5 | 44.5 | 77.1 |
| | SFT → DPO | $D_0 + D_1; D_2$ | 87.3 | **85.9** | **91.2** | 78.9 | **45.9** | 77.8 |
| Llama3$_{instruct}$ | SFT | $D_0 + D_1$ | 87.6 | 85.0 | 89.0 | 76.6 | 41.8 | 76.0 |
| | SFT → ORPO | $D_0 + D_1; D_2$ | 88.0 | 86.4 | 91.8 | 76.4 | 42.1 | 77.0 |
| | SFT → DPO | $D_0 + D_1; D_2$ | **89.0** | **86.8** | 90.8 | 80.5 | **41.9** | 77.8 |

$$L_{\text{ORPO}}(\pi_{\theta_{k+1}}) = - \lambda \cdot \log \sigma \left( \log \frac{\pi_{\theta_{k+1}}(y_w \mid x)}{1 - \pi_{\theta_{k+1}}(y_w \mid x)} \right.$$
$$\left. - \log \frac{\pi_{\theta_{k+1}}(y_l \mid x)}{1 - \pi_{\theta_{k+1}}(y_l \mid x)} \right) \tag{7}$$
$$- \log \pi_{\theta_{k+1}}(y_w \mid x)$$

## A.6 SELF-GENERATED PREFERENCE DATA

Existing preference datasets (Cui et al., 2023; Weyssow et al., 2024) do not include many these kinds of inter-disciplinary annotations. We use the current policy to generate preference data. Examples are provided in Table 18 and Table 19.

Table 18: Examples of self-generated preference pairs based on in-house web QA data (I). Texts have been translated into English.

| | |
|---|---|
| **question** | The perimeter of an isosceles triangle is 20, and the ratio of the lengths of the two equal sides is 1:2. What is the length of the base? A.10 B.4 C.4 or 10 D.5 or 8 |
| **reference answer** | B |
| **winning code (w)** | |

```
from sympy import symbols, Eq, solve

# Define the unknown
a, b = symbols('a b')

# Establish the equation
# The hypotenuse is twice the base
equation1 = Eq(a, 2*b)

# The perimeter is 20
equation2 = Eq(2*a + b, 20)

# Solve the equation
solution = solve((equation1, equation2), (a, b), dict=True)
solution
```

| | |
|---|---|
| **code result of w** | [a: 8, b: 4] |
| **losing code (l)** | |

```
from sympy import symbols, Eq, solve

# Define the unknown variable
a = symbols('a')

# Establish the equation
equation = Eq(2 * a + 2 * (2 * a), 20)

# Solve the equation
solution = solve(equation, dict=True)
solution
```

| | |
|---|---|
| **code result of l** | [a: 10/3] |

## A.7 OTHER DISCUSSIONS

**Language Diversity of Resources**: in this paper, we focus on large-scale question-answer pairs from educational websites for Chinese, and accordingly, our critic model used for guiding self-improvement is trained on Chinese data. While considering resources in other languages such as English could enhance the the generalizability of LLMs, it would require extensive human efforts for data collection and cleaning, which is beyond the scope of this work. On the other hand, since the backbone LLMs are pre-trained and aligned on multi-lingual data and our seed data includes English instruction data, the initial policy already exhibits reasonable performance on in-distribution (Section 3.3) and out-of-distribution benchmarks (Section 3.6). Self-improving this initial policy model on Chinese data may even improve its performance on English tasks. Finally, experiments show that the critic model is as effective at rating English responses as rating Chinese ones (Section 3.5).

**LLM Scalings**: Due to limited computational resources, our experiments focus on 7-8B LLMs. Generally, improving the math reasoning abilities of relatively small LLMs requires a large amount of training data and knowledge distillation (Li et al., 2024; Shao et al., 2024), which may not be necessary for larger LLMs.

Table 19: Examples of self-generated preference pairs based on in-house web QA data (II). Texts have been translated into English.

| question | Given that $a$ and $b$ are opposites, $c$ and $d$ are reciprocals, and the square of $x$ equals 4, try to find the value of the expression: $$x^2 - (a + b + cd)x + (a + b)^{2009} + (-cd)^{2008}.$$ |
| --- | --- |
| reference answer | 3 or 7 |
| winning code (w) | |

```
# Based on the analysis, first solve for the two possible values of
    x
# Since x squared equals 4, solve for x
x1 = 2
x2 = -2

# Next, calculate the value of the expression based on the value of
    x
# After simplification, the expression is x^2 - x + 1
# Calculate the value of the expression for x1 and x2
result1 = x1**2 - x1 + 1
result2 = x2**2 - x2 + 1

# Return results
{'x1 result': result1, 'x2 result': result2}
```

| code result of w | {'x1': 3, 'x2': 7} |
| --- | --- |
| losing code (l) | |

```
# Based on analysis, we have obtained the simplified form of the
    expression

# Define variables
# The value of a + b, since a and b are opposites, take a = 1, then
    b = -1
a_b_sum = 1
# The value of cd, since c and d are reciprocals, take c = 1, then
    d = 1
cd_product = 1
# The square of x
x_squared = 4

# Calculate the value of the expression
# x^2 - (a+b+cd)x + (a+b)^2009 + (-cd)^2008
result = x_squared - (a_b_sum + cd_product) * 2 + a_b_sum**2009 +
    (-cd_product)**2008

# Return result
{"result": result}
```

| code result of l | {2} |
| --- | --- |

1026
1027
1028
1029
1030
1031

Table 20: Examples of self-generated preference pairs based on WebInstruct (I). Texts have been translated into English.

| | |
|---|---|
| **question** | How do you find the indefinite integral of $\frac{5x^3-42x^2+73x-27}{(x-5)^2(x^2+4)}$ ? |
| **reference answer** | We can use partial fraction decomposition to rewrite the integrand as:\n\n(5x^3-42x^2+73x-27)/((x-5)^2(x^2+4)) = A/(x-5) + B/(x-5)^2 + (Cx+D)/(x^2+4)\n\nMultiplying both sides by the denominator, we get:\n\n5x^3-42x^2+73x-27 = A(x-5)(x^2+4) + B(x^2+4) + (Cx+D)(x-5)^2\n\nExpanding and equating coefficients, we get the following system of equations:\n\nA + C = 5\n-5A + B - 10C + D = -42\n4A + 25C - 10D = 73\n-20A + 4B + 25D = -27\n\nSolving this system, we find that:\n\nA = 2\nB = -3\nC = 3\nD = 1\n\nTherefore, the indefinite integral of (5x^3-42x^2+73x-27)/((x-5)^2(x^2+4)) is:\n\n∫ (5x^3-42x^2+73x-27)/((x-5)^2(x^2+4)) dx =\n\n∫ (2/(x-5) - 3/(x-5)^2 + (3x+1)/(x^2+4)) dx =\n\n2ln\|x-5\| + 3/(x-5) + 3/2ln(x^2+4) + 1/2arctan(x/2) + C\n####\nThe answer is 2ln\|x-5\| + 3/(x-5) + 3/2ln(x^2+4) + 1/2arctan(x/2) + C} |

**winning code (w)**

```
from sympy import symbols, integrate, factor

# Define the variable
x = symbols('x')

# Expressions for the numerator and denominator
numerator = 5*x**3 - 42*x**2 + 73*x - 27
denominator = (x - 5)**2 * (x**2 + 4)

# Factor the denominator
factored_denominator = factor(denominator)

# Calculate the definite integral
integral_result = integrate(numerator / factored_denominator, x)
integral_result
```

**code result of w**  $2*\log(x - 5) + 3*\log(x**2 + 4)/2 + atan(x/2)/2 + 3/(x - 5)$

**losing code (l)**

```
from sympy import symbols, integrate

# Define the variable x
x = symbols('x')

# Define the numerator and denominator
numerator = 5*x**3 - 42*x**2 + 73*x - 27
denominator = (x - 5)**2 * (x**2 + 4)

# Calculate the derivatives of the numerator and denominator
numerator_derivative = numerator.diff(x)
denominator_derivative = denominator.diff(x)

# Calculate the derivative of the original function
F_prime = numerator_derivative / denominator_derivative

# Use the Newton-Leibniz formula to compute the original function
F = integrate(F_prime, x)

# Return the result
{"F": F}
```

**code result of l**  {'F':  14*log(x  -  5)/29  +  379*log(x**2  -  5*x/2  +  2)/232 + 291*sqrt(7)*atan(4*sqrt(7)*x/7 - 5*sqrt(7)/7)/812}

1077
1078
1079

Table 21: Examples of self-generated preference pairs based on WebInstruct (II). Texts have been translated into English.

| question | What is the temperature in Fahrenheit if it is 25°C? |
|---|---|
| reference answer | |

```
\#\{:\textbackslash n(,"Fahrenheit","Celsius"),\textbackslash n("
    boiling point",212\^{}@F,100\^{}@C),\textbackslash n("freezing
     point",32\^{}@F,0\^{}@C),\textbackslash n("difference", 180F
    \^{}@,100C\^{}@)\textbackslash n:\}\#\textbackslash nSo\
    textbackslash n\#color(white) ("XXX")180 F\^{}@=100C\^{}@\
    textbackslash n\#color(white)("XXX")1C\^{}@=180/100F\^{}@\
    textbackslash n\#25\^{}@C\textbackslash n\#color(white)("XXX")
    \# is \#0\^{}@C + 25\^{}@C\textbackslash n\#color(white)("XXXX
    ")=25\^{}@C\# above freezing.\textbackslash n\#25C\^{}@ = 25
    xx180/100F\^{}@=45F\^{}@\#\textbackslash n\#45F\^{}@\# above
    freezing is \#32\^{}@F + 45\^{}@F = \textcolor{green}{77\^{}@F
    }\#}
```

**winning code (w)**

```
from sympy import symbols, Eq, solve

# Define the unknown variable
f = symbols('f')

# Given temperature in Celsius
c = 25

# Conversion formula
equation = Eq(f, c * 9/5 + 32)

# Solve the equation
solution = solve(equation, f, dict=True)
solution
```

| code result of w | [{f: 77.0000000000000}] |
|---|---|
| **losing code (l)** | |

```
# Define a function to convert degrees to minutes
def degrees_to_minutes(degrees):
    return degrees * 60

# Given degrees
degrees_celsius = 25

# Convert degrees to minutes
minutes = degrees_to_minutes(degrees_celsius)

# Calculate the corresponding temperature (Fahrenheit), since 1
    degree = 60 minutes, divide by 60
temperature_fahrenheit = minutes / 60

# Return the result
result = {'temperature_fahrenheit': temperature_fahrenheit}
result
```

| code result of l | {'temperature_fahrenheit': 25.0} |
|---|---|

