# OpenReview forum: "SIaM: Self-Improving Code-Assisted Mathematical Reasoning of Large Language Models"
_ICLR.cc/2025/Conference — ICLR 2025 Conference Withdrawn Submission_

### Official Review · Reviewer_z8vZ · 2024-10-31

**Soundness:** 3
**Presentation:** 2
**Contribution:** 2
**Rating:** 5
**Confidence:** 3

**Summary:**

The paper introduces a code-assisted self-improving method, which utilizes a critic model to verify the correctness of the implementation result of the code generated by the policy model, iteratively optimizing the policy model for large-scale web QA. The study evaluates a wide range of models and conducts extensive experiments on both in-domain and out-of-domain datasets, demonstrating the effectiveness of SIaM and its generalization capabilities.

**Strengths:**

1. The paper effectively employs an iterative refinement technique, utilizing a substantial number of QA pairs to enhance the mathematical reasoning capabilities of large language models.

2. The main experiments conducted are solid, with validation across multiple datasets and models that support the reliability of the findings.

3. The proposed method exhibits strong scalability, indicating its potential for broader applications in the field.

**Weaknesses:**

1. Despite presenting a practical framework for model training, the paper's novelty is somewhat limited. The methodology closely resembles existing approaches such as RFT, STAR, and V-STAR, with the primary distinction being its application to code-augmented COT.

2. Additionally, The motivation behind the study is not sufficiently compelling. The results seem somewhat predictable; adding TIR to traditional COT is generally expected to enhance performance. It would be beneficial for the authors to clearly explain the specific advantages of their code-augmented COT over traditional COTs with RFT. Consider additional experiments that could highlight these differences.

3. Given that the critic model is trained using GPT-4, the training labels are not guaranteed to be correct, which may affect the performance of the critic model. The authors should further clarify how their approach compares to exact match metrics to substantiate the reliability of the critic model's evaluations.

**Questions:**

1. The generalization of this paper largely relies on data from other domains, such as APE310K and CM, which also require golden labels. I believe a fairer comparison would involve training traditional methods like RFT or V-STAR on these datasets as baselines before comparing them to the proposed SIaM.

2. Regarding the advantages of RFT within the TIR context, it is unclear what specific improvements are achieved over traditional RFT. What is the rationale for utilizing TIR, and what unique benefits arise from the integration of these two methods? Additionally, Tool-Augmented approaches may focus more on achieving correct results rather than detailing the reasoning process. Is there a significant difference compared to directly using Chain-of-Thought (COT) techniques? I suggest conducting experiments to validate this.

3. For more complex reasoning tasks, such as MATH, any observed performance improvements appear marginal. The authors need to provide more comprehensive explanations rather than attributing these variations solely to evaluation scripts. Are there instances where code implementation might not be feasible? How might these situations limit the accuracy and scalability of the proposed methods?

---

### Official Review · Reviewer_dzq8 · 2024-11-01

**Soundness:** 2
**Presentation:** 2
**Contribution:** 2
**Rating:** 3
**Confidence:** 4

**Summary:**

The main purpose this paper is that previous methods for improving models to solve mathematical problems with code have largely relied on more powerful models to generate data and only used a limited number of datasets, such as GSM8k and MATH. This paper proposes using a critical model to compare the output of code execution with the golden answers. The amount of data used in this paper is at least an order of magnitude greater than that used in datasets like GSM8k and MATH.

**Strengths:**

Experiments are conducted both on English and Chinese

**Weaknesses:**

1. The paper lacks clarity overall, with the abstract and introduction failing to clearly explain the specific problem the authors are attempting to address.

2. Although the authors propose using a critic model to enhance the model's capabilities, they do not adequately justify the necessity of this approach. The primary purpose of incorporating code execution is to improve the model's ability to solve problems. However, Table 3 appears to lack a critical baseline—namely, the results of fine-tuning the model directly with the large amount of collected data. This omission leaves the purpose of this approach somewhat unclear.

3. The proposed method in the paper lacks innovation, appearing more as an experimental report rather than introducing any genuinely novel approach.

Grammars:
1. In Line 177 lacks definition of $j$

**Questions:**

N/A

---

### Official Review · Reviewer_ExDu · 2024-11-04

**Soundness:** 2
**Presentation:** 2
**Contribution:** 2
**Rating:** 5
**Confidence:** 5

**Summary:**

The paper introduces a novel approach to enhancing large language models (LLMs) for solving mathematical problems via coding. The authors propose using large-scale, expert-written, diverse math question-answer pairs. Their framework employs a code-based critic model to guide question-code dataset construction, quality control, and evaluation. They also explore alignment algorithms using self-generated instruction data to foster self-improvement.

**Strengths:**

- The authors propose a method to continuously self-improve LLMs by utilizing well-formatted math data from the web, focusing on the efficient comparison of reference answers and code execution results.

- A critic model is introduced to evaluate the performance of LLMs. This model is designed to handle diverse answer formats and improve the assessment of code responses, thereby enhancing the overall evaluation process.

**Weaknesses:**

- The self-improvement pipeline proposed in this paper is not particularly novel; it combines rejection sampling fine-tuning and DPO, mirroring approaches in [1] and [2], but with specific adaptations for code data. All these methods generate multiple model responses, selecting high-quality responses for SFT training data and constructing good-bad response pairs for DPO training data. While this paper generalizes these techniques to large-scale web QA pairs and code data for mathematical reasoning, it does not present any particularly new conclusions.

- The proposed method appears to be data-inefficient, utilizing over 1,000K samples yet producing suboptimal results. For instance, accuracy rates of 80.5% and 41.9% are reported on the MATH and GSM8K datasets, respectively. In contrast, other works, such as Dart-Math (which samples additional code responses from Deepseek-MATH-RL for difficult questions) [3], also using Llama3 as a base model, achieve accuracies of 81.1% and 46.6% with only 590K query-response pairs. The discrepancy in performance may be attributed to differences in response style, quality, and other factors.

[1] ChatGLM-Math: Improving Math Problem-Solving in Large Language Models with a Self-Critique Pipeline.

[2] ReFT: Reasoning with Reinforced Fine-Tuning.

[3] DART-Math: Difficulty-Aware Rejection Tuning for Mathematical Problem-Solving.

**Questions:**

- You may want to fix your typos Table '??' in A.4 in line 850.

- In Table 3, the scores for the initial model QWEN2Math_instruct on the English benchmarks are GSM8K: 79.5 and MATH: 48.0, which significantly differ from the official blog scores of GSM8K: 89.9 and MATH: 75.1. Could you please clarify the discrepancy between your reported scores and the official blog scores? Additionally, could you please give an explanation for why tools are indicated for QWEN2Math in the table, given its post-training with COT data?

---

### Note · Authors · 2024-12-16

I have read and agree with the venue's withdrawal policy on behalf of myself and my co-authors.